# Chemosensing Properties of Coumarin Derivatives: Promising Agents with Diverse Pharmacological Properties, Docking and DFT Investigation

**DOI:** 10.3390/molecules27185921

**Published:** 2022-09-12

**Authors:** Sadeq M. Al-Hazmy, Mohamed Oussama Zouaghi, Jamal N. Al-Johani, Youssef Arfaoui, Rania Al-Ashwal, Bechir Hammami, Ibrahim A. Alhagri, Nabil A. Alhemiary, Naceur Hamdi

**Affiliations:** 1Department of Chemistry, College of Science, Qassim University, Buraidah 51452, Saudi Arabia; 2Department of Chemistry, College of Science, Sana’a University, Sana’a P.O. Box 1247, Yemen; 3Laboratory of Characterizations, Applications & Modeling of Materials (LR18ES08), Department of Chemistry, Faculty of Sciences, University of Tunis El Manar, Tunis 2092, Tunisia; 4School of Biomedical Engineering and Health Sciences, Faculty of Engineering, Universiti Teknologi Malaysia, Johor Bahru 81310, Malaysia; 5Advanced Diagnostic and Progressive Human Care Research Group, School of Biomedical Engineering and Health Science Teknologi Malaysia, Johor Bahru 81310, Malaysia; 6Department of Chemistry, College of Science, Ibb University, Ibb P.O. Box 70270, Yemen; 7Department of Chemistry, College of Science and Arts at Sharurah, Najran University, P.O. Box 1988, Najran 11001, Saudi Arabia; 8Department of Chemistry, College of Science and Arts at Ar Rass, Qassim University, P.O. Box 53, Ar Rass 51921, Saudi Arabia; 9Research Laboratory of Environmental Sciences and Technologies (LR16ES09), Higher Institute of Environmental Sciences and Technology, University of Carthage, Hammam-Lif 1054, Tunisia

**Keywords:** coumarin, chemosensor, fluorescence, quantum yield, quenching, antibacterial, antioxidant, anti-inflammatory, antiproliferative, DFT investigation, docking simulation

## Abstract

In this work, a three-component reaction of 3-acetyl-4-hydroxycoumarine, malononitrile, or cyanoacetate in the presence of ammonium acetate was used to form coumarin derivatives. The chemical structures of new compounds were identified by ^1^H, ^13^C NMR and an elemental analysis. These compounds were examined in vitro for their antimicrobial activity against a panel of bacterial strains. In addition, these compounds were investigated for antioxidant activities by superoxideradical, DPPH (2,2-Diphenyl-1-picrylhydrazyl), and hydroxyl radical scavenging assays, in which most of them displayed significant antioxidant activities. Furthermore, these compounds were evaluated for anti-inflammatory activity by indirect hemolytic and lipoxygenase inhibition assays and revealed good activity. In addition, screening of the selected compounds **2**–**4** against colon carcinoma cell lines (HCT-116) and hepatocellular carcinoma cell lines (HepG-2) showed that that 2-amino-4-hydroxy-6-(4-hydroxy-2-oxo-2H-chromen-3-yl)nicotinonitrile **4** exhibited good cytotoxic activity against standard Vinblastine, while the other compounds exhibited moderate cytotoxic activity. Docking simulation showed that2-amino-4-hydroxy-6-(4-hydroxy-2-oxo-2H-chromen-3-yl)nicotinonitrile **4** is an effective inhibitor of the tumor protein HCT-116. A large fluorescence enhancement in a highly acidic medium was observed, and large fluorescence quenching by the addition of traces of Cu^2+^ and Ni^2+^ was also remarked.

## 1. Introduction

Among benzopyrones, coumarins are an important class found in many natural products and many pharmaceutically valuable compounds, and they possess various biological activities [1,2,3]. Natural and synthetic coumarin derivatives have attracted considerable attention due to their photochemotherapy and therapeutic applications in the treatment of cancer [4,5]. The reactions that play an increasingly essential role in combinatorial chemistry and drug discovery are the multi-component reactions that are highly valued as synthetic methodologies, because several elements of diversity can be introduced into a molecule in a single step [6]. First, DFT [7,8] study allows us to predict the local reactivity of the reagent’s sites and analyze the acidity and basicity of the molecule [9]. Moreover, TD-DFT theory [10,11,12] permits us to obtain UV spectra with the aim of evaluating the optical properties such as oscillator strength, fluorescence quantum yield, etc. [13]. In this way we have realized docking calculations to see the inhibiting behavior of the 2-amino-4-hydroxy-6-(4-hydroxy-2-oxo-2H-chromen-3-yl) nicotinonitrile **4** product against several proteins, and to provide detailed analyses of the anti-inflammatory and anti-proliferative biological activities of the titled organic product. An important sensor in chemical biology for the sensitive detection of metal ions in biomedical applications is the use of fluorescent probes as chemical sensors [14,15].

Furthermore, the detection and accurate estimation of radiation doses have become an important challenge facing those working in the field of detecting the presence of simple radiation doses in various applications. Low radiation doses in particular pose a real problem today due to the lack of appropriate dosimeters in this range [16,17]. Several forms of dosimeters that are suitable for different dose densities have been studied and made rapid progress, such as photographic film, the Geiger–Müller counter, and the proportional counter [17,18]. This paper aims to describe the pharmacological proprieties of coumarin derivatives. We alsopresent several theoretical studies such as an explanation of the tautomeric equilibrium and a mapped electrostatic potential surface analysis to better explain the reaction trend and the proposed reaction pattern. Molecular modeling studies are also performed to investigate the binding of 2-amino-4-hydroxy-6-(4-hydroxy-2-oxo-2H-chromen-3-yl)nicotinonitrile **4** with several proteins. There are also UV calculations to study the absorption spectra of compound **4** in different solvents and fluorescent activity. Furthermore, biological tests on the synthesized compounds: antibacterial activity, antioxidant activity, and anti-inflammatory activity were studied. In addition to this, compound **4** was also been explored for its ability to decrease fluorescence in the presence of certain ions, and thus exhibit chemosensing activity.

## 2. Experimental Section

### 2.1. General Information

The experiments were performed under Argon atmosphere. The MBRAUN SPS 800 solvent purification system (Malatya, Turkey) was used to purify the solvents. The ^1^H NMR and ^13^C NMR spectra were recorded at a frequency of 400 MHz and 100 MHz, respectively. The TMS was used as an internal standard for both ^1^H and ^13^C NMR. NMR analyses were executed in 5 mm NMR tubes. NMR multiplicities are listed as s = singlet, d = doublet, t = triplet and m = multiplet signals. IR spectra were recorded on a 398 spectrophotometer. The elemental microanalysis was performed on an ElementarVario El III Carlo Erba 1108 elemental analyzer (Langenselbold, Germany), and the values found were within ±0.4% of the theoretical values. The melting points were measured with the Kofler bench.

### 2.2. 3-Acetyl-4-hydroxycoumarine

The 4-hydroxy-2H-chromen-2-one (3 g, 1.86 mmol) was added to the phosphorus oxychloride (5.6 mL) in acetic acid (16 mL). The mixture was refluxed for 30 min. At the end of the reaction the mixture was cooled, and the precipitate was separated and recrystallized from ethanol to give the 3-acetyl-4-hydroxy-2Hchromen-2-one as white needles. Yield 2.7 g (90%); mp: 135 °C. IR spectrum, νcm^−1^: 3185 (OH); 1705 (CO); 1700 (O–CO lactone). ^1^H NMR spectrum (CDCl_3_). δ (ppm): 2.72 (3H, s, CH_3_); 7.98 (1H, s, H_5_); 7.95 (1H, dd, J = 7.8 H_z_); 7.1–7.4 (2H, m, H_6_, H_7_); 17.69 (1H, s, OH). ^13^C NMR spectrum (CDCl_3_), δ ppm: 29.9 (CH_3_); 178.5 (CO); 159.8 (C_4_); 154.6 (C_2_); 101.26 (C_3_); 115.2 (Carom.), 152.4 (Carom.), 122.1 (Carom.), 123.2 (Carom.), 126.4 (Carom.), 119.3 (Carom.).

### 2.3. General Procedure for the Synthesis of 4-aryl-1,2-dihydro-6-(4-hydroxy-2-oxo-2H-chromen-3-yl)-2-oxopyridin-3-carbonitrile 3

A solution of 3-Acetyl-4-hydroxycoumarin (0.5 g; 2.5 mmol) and the 3-hydroxybenzaldehyde (2.5 mmol) in the presence of ethyl cyanoacetate (0.26 mL; 2.5 mmol) and ammonium acetate (0.38 g; 5 mmol) was refluxed in 10 mL of DMC; the obtained solid was filtered off and recrystallized from methanol.

Yield: 82%; 0.76 g, mp: 236 °C. ^1^H NMR spectrum (DMSO-d_6_)δppmδ5.75 (s,1H, H_2′_); 7.78–9.77 (m, 8H, Ar-H); 9.77 (s,1H, NH). ^13^C NMR spectrum (DMSO-d_6_)δ (ppm) 54.9 (C_1′_); 117.4 (Carom.); 151.3 (Carom.); 123.4 (Carom.); 124.3 (Carom.); 127.2 (Carom.); 117.5 (Carom.); 140.1 (Carom.); 121.4 (Carom.); 130.1 (Carom.); 116.3 (Carom.); 158.2 (Carom.); 110.4 (Carom) (Carm.); 157.9 (C_4,5″_); 180.6 (C_2_); 191.3 (C_5′_). IR spectrum, ν cm^−1^: 3050 (O–H); 3350 (NH); 1715 (CO_lactone_); 1699 (CO amide) 1600 (C=C). Anal. Calcd for C_21_H_12_O_5_N_2_: C, 67.74%; H, 3.25%; N, 7.52%. Found: C, 67.8; H, 3.25; N, 7.5.

### 2.4. Preparation of 2-amino-4-hydroxy-6-(4-hydroxy-2-oxo-2H-chromen-3-yl)nicotinonitrile 4

To a mixture of 3-acetyl-4-hydroxycoumarin (1 g, 5 mmol) and the 3-hydroxybenzaldehyde (2.5 mmol) in ethyl alcohol (20 mL), malononitrile (0.33 g, 5 mmol) and ammonium acetate (0.75 g, 10 mmol) were added. The reaction mixture was refluxed for 6 h. The obtained solid was filtered off, washed with absolute ethyl alcohol and recrystallized from methyl alcohol to give the desired compounds.

Solid (Yield 95%), mp = 183 °C; IR: m 3368 (–NH_2_), 1716 (s) (lactone [C=O), 1577 (C=C), 1018(s) (sym) (C–O–C); ^1^H NMR: δ (ppm) 7.40 (s, 1H, Hethy), 7.96–8.20 (m, 4H, Ar–H), 13.2 (s, 1H, OH); ^13^C NMR (ppm): 158.2 (C_2_); 102.1 (C_3_); 149.2 (C_4_); 146.7 (C=N); 117.5 (Carom.), 151.4 (Carom.), 123.2 (Carom.), 125.3 (Carom.), 128.2 (Carom.), 116.5 (Carom.). Anal. Calcd for C_15_H_9_O_4_N_3_: C 61.02%; H 3.07%; N, 14.23%. Found: C, 61.01; H, 3.07; N, 14.25.

### 2.5. Antibacterial Activity

Bacterial strains, media, and growth conditions and minimum inhibitory concentration (MIC) were obtained according to our previous work [19,20,21].

### 2.6. Antioxidant Activity

1-diphenyl-2-picrylhydrazyl (α,α-diphenyl-β-picrylhydrazyl; DPPH) is characterized as a stable free radical by virtue of the delocalization of the spare electron over the molecule as a whole, so that the molecule does not dimerize, as would be the case with most other free radicals. The delocalization of electrons also gives rise to the deep violet color, which is characterized by an absorption band in ethanol solution centered at about 517 nm [22]. When a solution of DPPH is mixed with that of a substrate (AH) that can donate a hydrogen atom, this gives rise to the reduced form with the loss of this violet color. In order to evaluate the antioxidant potential through free radical scavenging by the test samples, the change in the optical density of DPPH radicals is monitored. The sample extract is diluted with methanol, and 2 mL of DPPH solution (0.5 mM) is added. After 30 min, the absorbance is measured at 517 nm. The percentage of the DPPH radical scavenging is calculated using the equation given below:Inhibition (%) = [(Absorbance Control − Absorbance sample)/Absorbance Control] × 100

### 2.7. Hydroxyl Radical Scavenging Assay

Hydroxyl radical is one of the potent reactive oxygen species in the biological system that reacts with the polyunsaturated fatty acid moieties of cell membrane phospholipids and causes damage to cells [19]. The reaction mixture (1.0 mL) consists of 100 μL of 2-deoxy-Dribose (28 mM in 20 mM of KH_2_PO_4_-KOH buffer, pH 7.4); 500 μL of the compound; 200 μL of EDTA (1.04 mM) and 200 μM of FeCl_3_ (1:1 *v*/*v*); 100 μL of H_2_O_2_ (1.0 mM); and 100 μL of ascorbic acid (1.0 mM), which is incubated at 37 °C for 1 h. One milliliter of thiobarbituric acid (1%) and 1.0 mL of trichloroacetic acid (2.8%) are added and incubated at 100 °C for 20 min. After cooling, absorbance is measured at 532 nm against a blank sample.

### 2.8. Anti-Inflammatory Activity

Lipoxygenase inhibition assay and indirect hemolytic assay were performed according to our previous work [23,24,25].

### 2.9. Photochemical Quantum Yields

The photochemical quantum yields of 2-amino-4-hydroxy-6-(4-hydroxy-2-oxo-2H-chromen-3-yl)nicotinonitrile **4** (φ_c_) were measured using the A. J. Lees method that considers the decrease in absorbance at the excitation wavelength as photo-irradiation proceeds [26].

### 2.10. Fluorescence Quantum Yields

Fluorescence quantum yields in liquids were determined using the optically dilute solution relative method with either 9,10-diphenyleanthracene or quinine sulphate solutions, depending on the emission wavelength range. The intensity of light was determined using ferrioxalate actinometry [27,28,29]. The following Equation (1) was applied to calculate the fluorescence quantum yields:(1)$$\varphi_{f}(s)=\varphi_{f}\left(r\right)\times \frac{\int I_{S}}{\int I_{r}}\times \frac{A_{r}}{A_{S}}\times \frac{n_{S}^{2}}{n_{r}^{2}}$$

The integrals denote the corrected fluorescence peak areas, A denotes the absorbance at the excitation wavelength, and n denotes the solvent’s refractive index. The subscripts s and r indicate sample and reference, respectively.

The FTIR spectra of the synthesized compound were recorded in the wavenumber range 500–4000 cm^−1^ using a Bruker FT-IR spectrophotometerby Shimadzu FTIR spectrometer (model 8000, made in Tokyo, Japan) and an Agilant spectrometer (carry 600 FTIR, Santa Clara, CA, USA). The range of measuring was from 400 to 4000 cm^−1^.

### 2.11. Computational Details

All calculations were carried out with the Gaussian A16 program [30] at the B3LYP functional [31,32,33] level of theory and the 6-31G+(d) basis set. We considered the effect of several solvents (acetic acid, water, chloroform, acetonitrile, THF and chloroform) using the CPCM solvation model [34,35,36,37].

In order to obtain UV absorption spectra, 30 of the lowest electronic excitation energies ($\Delta E_{0n}$) were computed using the TD-DFT [38,39] method at B3LYP/6-31+G(d) level. The optical UV absorption spectra were simulated by associating each transition with a Gaussian function that hada full width at the half maximum (FWHM) of 0.33 eV [38]. The emission UV spectra were computed based on the 30 lowest electronic transitions and considered the singlet–singlet transitions.

The computed spectrum can be shifted [39], since one of the purposes of this work is to compare absorption and emission band shapes between different solvents, as well as between simulations and experiment. When comparing simulations to experimental spectra, the energy shift was selected such that the absorption or emission maximum of the simulated spectrum would coincide with the experimental spectrum one, and this is indicated by “shifted” in the x-axis label. Moreover, there is no shift, or rather the $\Delta {Ε}_{0-0}$ transition was arbitrarily set equal to the vertical excitation energy when comparing the simulated spectra.

### 2.12. Molecular Docking

We performed a molecular docking simulation intending to predict the best binding configuration of a ligand to a macromolecular partner and for a detailed investigation of the interactions between the organic synthesized product and the protein amino-acids. It generates many possible positions of the ligand within the protein-binding site. The best pose of the ligand was selected based on its best conformation, providing the lowest free binding energy. The high-resolution crystallographic structures of enzymes (co-crystallized with either inhibitors or natural substrates) were downloaded from the Research Collaborator for Structural Bioinformatics, RCSB database, as indicated in the following link https://www.rcsb.org/pages/policies (accessed on 9 May 2022). The downloaded structures represent either survival or virulence-determining enzymes within each investigated bacterial cell.

Docking studies were carried out using the ezCADD software [40]. The “fpocket3” cavity detection [41] and Smina program [42] were used in this study. We cleaned the protein by using PyMOL software (The PyMOL Molecular Graphics System, v1.6-alpha; Schrodinger LLC, New York, NY, USA, 2013). The obtained results are visualized using the Discovery Studio 2016 Client software.

### 2.13. Preparation of PVA/DHCOC Nanocomposite Films

The thin films-based PVA polymer mixed with various amounts of 2-amino-4-hydroxy-6-(4-hydroxy-2-oxo-2H-chromen-3-yl)nicotinonitrile **4** filler was developed following a previously reported casting method [43].

## 3. Results and Discussion

In continuation of our previous work [44,45,46], to synthesize bioactive heterocyclic compounds under mild conditions, herein, we wish to report a mild and efficient procedure for the synthesis of some coumarins derivatives via the one-pot, three-component reaction of 3-acetyl-4-hydroxy coumarin with malononitrile or ethyl cyanoacetate in the presence of ammonium acetate. The development of all reactions was tracked by TLC (Figure 1).

We employed the commercially available 4-hydroxycoumarin as the starting material for synthesizing the title compound. The 3-acetyl-4-hydroxycoumarin precursor was prepared by a Knoevenagel reaction in basic conditions with a 90% yield. For the acetylation of the 4-hydroxycoumarin to give 3-acetyl-4-hydroxycoumarin **2**, the method of [47] was employed using glacial acetic acid as an acetylating agent in the presence of POCl_3_. The reaction was rapid, without competition from the intramolecular condensation of 4-hydroxycoumarin [48,49,50] (Figure 1).

Due to the exceptional reactivity of the acetyl group in 3-acetyl-4-hydroxycoumarin, as well as the versatile biological activities of coumarin derivatives, thereafter, compound **4** was obtained from 3-acetyl-4-hydroxycoumarin and 3-hydroxybenzaldehyde by a multicomponent reaction. The target 2-amino-4-hydroxy-6-(4-hydroxy-2-oxo-2H-chromen-3-yl)nicotinonitrile **4** was obtained in a good yield (95%), and then we decided to extend our three-component synthesis reaction in order to obtain compound **3**. The reaction of 3-acetyl-4-hydroxycoumarin with cyanoacetate proceeded extremely rapidly. The elemental analysis and spectroscopic data of the obtained products **3**–**4** supported the assigned structures. The IR spectrum of **3** exhibits two strong stretching frequencies in the regions of 3350 and 1750 cm^−1^, which are attributable to the NH and C=O groups, respectively. Its ^1^H-NMR spectrum displayed five singlet signals for the H_2′_, in addition to the characteristic multiplet signal for the eight aromatic protons. Moreover, its ^13^C-NMR showed signals at 54.9, 180.6, and 119.0 ppm, which corresponds, respectively, to C_1′_, C_2_, and CN. Aromatic signals between 114.3 and 136.4 ppm, and the signals at 100.9 ppm and 191.3 ppm are related to C_3_ and C_5′_, respectively. In IR spectra, 2-amino-4-hydroxy-6-(4-hydroxy-2-oxo-2H-chromen-3-yl)nicotinonitrile **4** showed a very strong band at 1683 cm^−1^ for the carbonyl (C=O) stretching of the δ-lactone ring that is present in the coumarin nucleus. The strong bands for C–O and C=N stretching vibrations were observed at 1714 and 1608 cm^−1^, respectively. The aromatic C–H stretching vibrations were observed between 3093 and 3144 cm^−1^. The δ-lactone carbonyl stretching frequency was observed somewhat lower than the expected frequency (1720 cm^−1^). In the ^1^H-NMR spectra of the 2-amino-4-hydroxy-6-(4-hydroxy-2-oxo-2H-chromen-3-yl)nicotinonitrile **4**, the aromatic proton signals were observed in the region of δ6.89–8.11 ppm, while the C–H and signal was observed at 6.89 ppm. This proton signal shift in the downfield region is due to the peri effect of the nitrogen atom. 2-amino-4-hydroxy-6-(4-hydroxy-2-oxo-2H-chromen-3-yl)nicotinonitrile **4** showed signals at expected δ values in the ^13^C-NMR spectra. The aromatic carbon signals were observed between δ116.7 and 136.8 ppm, while the δ-lactone carbonyl carbon C_2_ appeared at δ181.7. The acetate ion deprotects 3-acetyl-4-hydroxycoumarin; the latter makes an attack on the methylene of the malononitrile, which leads to the formation of an intermediate (**1**). Then, a rearrangement is made to obtain an intermediate (**2**), which, by nucleophilic attack of malononitrile leads to intermediate (**3**), which follows a succession of reactions to lead to the final 2-amino-4-hydroxy-6-(4-hydroxy-2-oxo-2H-chromen-3-yl)nicotinonitrile **4** (Figure 2).

### 3.1. Theoretical Results

#### 3.1.1. Modeling of the Product

The structures of the reagent and product were optimized at B_3_LYP/6-31+G(d) level (Figure 1). In order to search for the most stable product structure, four scan calculations were performed. The most stable one is indicated in Figure 2 (green curve).

Figure 3 shows that the HOMO is localized mostly in the center of the 2-amino-4-hydroxy-6-(4-hydroxy-2-oxo-2H-chromen-3-yl)nicotinonitrile **4**, whereas the LUMO is shifted to the acetyl-hydroxycoumarin moiety.

#### 3.1.2. Study of the Tautomeric Equilibrium (Imine↔Amine)

The tautomeric equilibrium (Amine↔Alcohol), as indicated in Figure 4, was studied. To search for the most stable tautomer, the transition state TS was computed, followed by intrinsic reaction coordinates (IRC), in order to confirm the reliability of the TS for both desired tautomers. We compared the activation energy $E_{a}$ and free energy of reaction values ($\Delta_{r}E$) in the gas phase, explicit model and IEF-PCM implicit solvation model with water as a solvent. The computed IRC plots are represented in Figure 5 and the obtained data are listed in Table 1.

The examination of the IRC plot shows that the amine tautomer is more stable than the alcohol one, with a free energy reaching −7.72 and −6.86 kcal·mol^−1^ in the case of the implicit model and gas phase, respectively, whereas this value is −1.65 kcal.mol^−1^ for the explicit model. The highest value of the activation energy can be explained by the stability of the amine tautomer in the gas phase or in the implicit model. On the contrary, in the case of the explicit model, the activation energy is equal to 14.18 kcal·mol^−1^, indicating the possibility of the equilibrium between both forms. We can conclude that the amine is the most stable tautomer and the explicit solvation model is suitable to investigate this kind of tautomeric equilibrium—see Figure 5. The impact of water as a suitable solvent in order to obtain the lowest activation energy values in the case of similar tautomeric equilibria was studied previously, and this is in good agreement with our obtained results [51].

#### 3.1.3. Mapped Electrostatic Potential Surface (MEPs) Analysis

In order to investigate the local reactivity of the reagent, we analyzed the maximum and minimum potential $V_{s}$ values deduced from the MEP surfaces (Table 2). The positive regions of MEP shown in blue correspond to electrophilic reactivity, and the negative regions shown in red are responsible for nucleophilic reactivity. From the MEPs shown in Table 2, it is obvious that there is an extensive region of positive electrostatic potential around the methyl group, which leads us to conclude that the methyl protons are electrophilic sites and, consequently, the most acidic ones that underwent a nucleophilic attack by acetate ammonium base, whereas the regions around ketone (C_13_=O_23_), cyclic ketone (C_8_=O_12_) and alcohol group have negative electrostatic potential and may act as nucleophilic sites. We have found that H_21_(methyl proton) possesses the highest $V_{S,max}$ value and, consequently, the most mobile proton. The obtained result confirms the proposal mechanism sketched in Figure 2. The maximum and minimum potential values of the desired product are also summarized in Table 3. As relevant results, we found that H_40_ is the most mobile hydrogen and N_28_ and O_19_ are the most nucleophilic sites.

### 3.2. UV Calculations

#### 3.2.1. TD-DFT Absorption UV Spectra Analysis

The UV absorption spectra were then computed for the 2-amino-4-hydroxy-6-(4-hydroxy-2-oxo-2H-chromen-3-yl)nicotinonitrile **4** as a function of different solvents (water, ethanol, chloroform, acetonitrile, THF and acetic acid). The obtained simulated spectra are presented in Figure 6 and the optical characteristics are listed in Table 4.

Table 4 shows that the most favorable transitions of the maximum wavelength at 290–340 nm are related to the frontier MOs HOMO→LUMO or HOMO-1→LUMO. Furthermore, the optimal hyperchromic shift (jump of ε and oscillator strength values) is observed using water and acetonitrile as solvents.

#### 3.2.2. TD-DFT Fluorescence UV Spectra Analysis

The emission (fluorescence) UV spectra were calculated in order to predict the fluorescence quantum yield and the optical energy gap. In this study, only singlet–singlet transitions are taken into account. The calculated fluorescence UV spectra are indicated in Figure 7. To compute the optical energy gap, we searched for the intersection (absorption–emission) wavelength as a common method. The obtained optical energy gap ($eg_{opt}$) is summarized in Table 5. We found that THF and acetic acid give the lowest $eg_{opt}$ values. Moreover, we predicted the fluorescence quantum yield (${ɸ}_{f}$) by using spectral analysis software. According to the data in Table 6, the optimal ${ɸ}_{f}$ value is revealed in the case of ethanol, water, and acetonitrile solvents.

### 3.3. UV-Visible and Fluorescence Spectra of 2-amino-4-hydroxy-6-(4-hydroxy-2-oxo-2H-chromen-3-yl)nicotinonitrile4

The spectroscopic and photophysical characteristics of 2-amino-4-hydroxy-6-(4-hydroxy-2-oxo-2H-chromen-3-yl)nicotinonitrile **4** in different solvents that were selected based on their ability to dissolve the dye, as well as a difference in polarity, are summarized in Table 7. From Table 7, the Stokes shift increases as the relative polarity of the solvent increases, marking an increase in the dipole moment upon excitation. This data can again be used to indicate charge transfer transitions. The observed fluctuations in dipole moments can be explained by the resonance structures of the dipoles. 2-amino-4-hydroxy-6-(4-hydroxy-2-oxo-2H-chromen-3-yl)nicotinonitrile **4** has a bigger excited state dipole moment (µe) than it does a ground state dipole moment (µg). Figure 8 shows the absorption spectra of 2-amino-4-hydroxy-6-(4-hydroxy-2-oxo-2H-chromen-3-yl)nicotinonitrile 4 of a concentration of 7.6 × 10^−5^ M in solvents withdifferent polarities. The increase in solvent polarity shows a considerable influence on wavelength maxima, electronic absorption and emission. As a significant result, we found that the UV-vis absorption theoretical spectra shape is similar to the experimental one (Figure 8). We can conclude that the selected theoretical method is reliable. The red shifts (bathochromic) in the absorption and fluorescence maximum with an increase in the polarity of the solvent indicate a decrease in the ground state (S_0_) dipole moment of the dye molecule rather than the excitation state (S_1_) [52,53], as shown in Figure 9, Figure 10, Figure 11, Figure 12 and Figure 13. The UV-visible absorption spectra of coumarin derivative dyes obtained at room temperature (see Figure 9) (7.6 × 10^−5^ M) in chloroform show a clear strong absorbance at 240 nm due to spin-allowed S_0_→S_2_ transition, in addition to a low energy peak at 309 nm, which is lower in intensity than the previous one and indicates a spin-allowed S_0_→S_1_ transition [54].

In this study, we observed that the occurrence of absorption bands in the regions of 240–260 nm and 307–316 nm are due to π→π* transitions. Both longer and shorter wavelength show about a 10-nm wavelength variation due to an increase in the polarity of the solvents.

Figure 14 and Figure 15 show the tangible influences of solvent polarity on the position of the λ_ab_ max. of the absorption and emission spectra. The hypsochromic band shifts (blue shift) in the electronic absorption and emission spectra of 2-amino-4-hydroxy-6-(4-hydroxy-2-oxo-2H-chromen-3-yl)nicotinonitrile **4** with the increasing solvent polarity of the medium is expected and reflects a decrease in the ground state dipole moment of the dye molecule upon excitation, and an increase in its ground-state dipole moment as a result of solvent polarization. This significant effect of the solvent relative polarity on the absorption peak of the studied dye was observed starting from 307 nm in acetonitrile (CH_3_CN) with a relatively high intensity, up to 316 in carbon tetrachloride, as shown in Figure 12 and Table 7.

In Figure 14, it was noticed from the absorption spectra that there is a relative difference in the absorption density of the π-π* and n-π* transitions of the studied dye in both chloroform and ethanol solvents, while the density of π-π* and n-π* transitions absorption peaks is approximately equaled in the solid matrix. Therefore, we can suggest that the interaction of the complex with the polymer contributes to a better overlap between the frontier orbitals of our complex, thus, leading to an enhancement in absorbance at a longer wavelength.

As illustrated in Figure 15, the Stokes shift increases as the relative polarity of the solvent increases, marking an increase in the dipole moment upon excitation. This data can be again used to indicate charge transfer transitions. The observed fluctuations in dipole moments can be explained by the resonance structures of the dipoles [55,56]. 2-amino-4-hydroxy-6-(4-hydroxy-2-oxo-2H-chromen-3-yl)nicotinonitrile 4 has a bigger excited state dipole moment (µe) than it does a ground state dipole moment (µg). This observed high Stokes shift of the dye in most solvents, as shown in Figure 10, which ranges from 42 nm in non-polar solvents to 102 nm in polar solvents, is a sign of promising dye and excluding the occurrence of photons re-absorption, which improves its efficiency. This property was enhanced in the solid matrices (2-amino-4-hydroxy-6-(4-hydroxy-2-oxo-2H-chromen-3-yl)nicotinonitrile 4 doped PVA), where it rose to 112 when excited at a wavelength of 365 nm thin film), and about 120 when excited at 376 nm.

A solution of 2-amino-4-hydroxy-6-(4-hydroxy-2-oxo-2H-chromen-3-yl)nicotinonitrile **4** was studied at a small concentration of 6.5 × 10^−5^ M to reduce the self-absorption phenomena. Figure 16 and Figure 17 show the decrease in emission intensity in different solvents and the decrease in the fluorescence quantum yield ɸ_f_ with the increase in solvent polarity [57,58,59,60].

### 3.4. Photostability

The quantum yield of the photochemical transformation (Φ_c_) of 2-amino-4-hydroxy-6-(4-hydroxy-2-oxo-2H-chromen-3-yl)nicotinonitrile **4** was measured in CCl_4_, ethanol and for coumarin 4-doped PVA. The values of Φ_c_ were found to be 2 × 10^−3^, 3.3 × 10^−3^, and 1 × 10^−3^, respectively, indicating more stability in the solid matrix due to the restriction of the molecules. Figure 18 shows the decrease in the emission intensity of 2-amino-4-hydroxy-6-(4-hydroxy-2-oxo-2H-chromen-3-yl)nicotinonitrile **4** upon irradiation using 365 nm.

The presence of the electron-donating OH group in position 4 of 2-amino-4-hydroxy-6-(4-hydroxy-2-oxo-2H-chromen-3-yl)nicotinonitrile **4** increases the charge transfers character of the transition. The phenolate species that forms in basic solution (EtOH/NaOH) has two maximum peaks at 222 and 308 nm, red-shifted relative to 2-amino-4-hydroxy-6-(4-hydroxy-2-oxo-2H-chromen-3-yl)nicotinonitrile **4** in acidic medium (215 and 301 nm) [61]. No significant effect was observed on the emission maximum when alkali was added; only a blue shift in the emission maxima was observed from 410 in the neutral and basic medium to 333 nm in the re-acidified medium by adding sulfuric acid—see Figure 19 and Figure 20.

### 3.5. Effect of Viscosity of the Medium

The fluorescence intensity of 2-amino-4-hydroxy-6-(4-hydroxy-2-oxo-2H-chromen-3-yl)nicotinonitrile **4** increases with increasing the proportion of glycerol in the glycerol–ethanol mixture. Figure 21 shows that the fluorescence quantum yield in glycerol is doubled compared to that in non-viscous ethanol [62,63].

### 3.6. Biological Activities

#### 3.6.1. Antibacterial Activities

Compounds **2**–**4** were evaluated for in vitro antibacterial and antifungal activities against various Gram-positive, Gram-negative bacteria and fungal species. The results are shown in Table 8 and Table 9. Standard antibacterial **AMC** was also tested for comparison with synthesized compounds **2**–**4**.

The minimum inhibitory concentration (MIC) against the above organisms was determined by the method of dilutions [64]. The results are given in Table 9.

#### 3.6.2. Antioxidant Activity

The synthesized compounds **2**–**4** were tested for in vitro antioxidant activity by DPPH radical, hydroxyl radical and superoxide radical scavenging assays. The IC50 values of the standards and test samples are summarized in Table 10.

#### 3.6.3. Anti-Inflammatory Activity

We next examined the anti-inflammatory activities of the synthesized compounds **2**–**4** by lipoxygenase inhibition and phospholipase A_2_ (PLA2) inhibition assays. The IC50 values of the standards and test samples in both assays are given in Table 11. In both the assays, the synthesized compounds **2**–**4** showed potent anti-inflammatory activity in lipoxygenase inhibition assay (7.3–8.5 µM) and PLA2 inhibition assay (1004–1110 µM). It should be noted that **2** and **4** nearly have anti-inflammatory activities, as do that of standards indomethacin and aristolochic acid in lipoxygenase inhibition assay.

#### 3.6.4. Antiproliferative Activity

Screening of the selected compounds (**2**–**4**) against human colon carcinoma cancer cell lines and hepatocellular carcinoma cells lines were tested for potential cytotoxicity using the Mossman [65], Gangadevi and Muthumary [66] methods, which revealed that the compound **2** had IC_50_ (7.76 and 11.75 µg) in both human colon carcinoma cancer cell lines and hepatocellular carcinoma cells lines, respectively. Results are given in Table 12.

### 3.7. Docking Result Analysis

Molecular modeling studies were performed to investigate the possible binding mode of 2-amino-4-hydroxy-6-(4-hydroxy-2-oxo-2H-chromen-3-yl)nicotinonitrile 4 product targeting the crystal structures of the listed proteins in Table 13.

First, we searched for the number of possible cavities (n) that may be inhibited by the 2-amino-4-hydroxy-6-(4-hydroxy-2-oxo-2H-chromen-3-yl)nicotinonitrile **4** and their associated volumes. Second, the best cavity was selected followed by the simulation of the potential organic ligand poses in interaction with the protein. Based on the binding free energy, we analyzed the organic molecule behavior within the protein crystal structure. The obtained results are summarized in Table 14.

According to the data in Table 15, we presented the best poses of the incorporated inhibitor within the listed protein—Figures a in Table 15. However, the interactions between the organic compound 2-amino-4-hydroxy-6-(4-hydroxy-2-oxo-2H-chromen-3-yl)nicotinonitrile **4** and the seven proteins are sketched in Figures b in Table 15.

The analysis of 2D diagrams (Figures b in Table 15) shows that the aromatic rings and hetero-atoms of the 2-amino-4-hydroxy-6-(4-hydroxy-2-oxo-2H-chromen-3-yl)nicotinonitrile **4** molecule have important impacts on the interaction with the protein’s amino acid. The coumarin4 aromatic rings interact with the amino acids via π-alkyle, π-π T-shaped, π-cation, π-anion, π-σ and π-donor hydrogen interactions.

In the case of the anti-inflammatory activity, we observed that the interaction with the lipoxygenase exhibits the lowest binding free energy value (−10.3 kcal·mol^−1^) in the case of the 6N2W protein. In the case of 4NRE, however, we discovered more interactions between the compound **4** and amino acids. Moreover, coumarin 4 nitrogens and oxgens interact with 4NRE via conventional hydrogen bonds and unfavorable donor–donor interactions. The histidine as an amino acid interacts with the coumarin4 ketones in the case of both proteins (6NRE and 4NRE). For PLA_2_, we investigated the interactions between coumarin4 and three proteins (1TH6, 2QU9 and 4DBK). Aside from the interactions between the amino acids and the coumarin4 aromatic rings, we observed conventional hydrogen bonds with the ketones in the cases of 2QU9 and 4DBK. Furthermore, there is no interaction with the coumarin4 nitrogen and only the aromatic rings are involved in the amino acid-coumarin **4** interaction in the case of 1TH6.

In the case of the antiproliferative activity, we studied the interactions of coumarin4 with HepG-2 (2W3L) and HCT-116 (1YWN) proteins. We found that coumarin4 is an effective inhibitor of a novel tumor (colon carcinoma HCT-116) and also of hepatocellular carcinoma cell lines (HepG-2) based on the obtained binding free energy values and several interactions between the ligand (coumarin4) and the many amino acids (VAL, LEU, CYS, ASN, ARG, PHE, ALA, GLY and TYR). This outcome is in good agreement with the IC_50_ values in the experimental section. We can conclude that the docking simulation enabled us to determine the contribution of heteroatoms (oxygen and nitrogen) and aromatic rings as a conjugated system to improving the biological activities of coumarin4 as an inhibitor for the studied proteins.

### 3.8. Coumarin 4 as a Chemosensor for Cu^2+^ and Ni^+2^ Ions

Figure 22 and Figure 23 show a major quenching in the fluorescence intensity in the presence of Cu^2+^ ions and Ni^+2^, respectively, where the successive increase in the copper ion concentration clearly results in a decrease in the emission intensity at the two emission maxima (436 and 411); there was even a sharp decrease with the first addition of a small amount of Cu^2+^ ions. This apparent quenching indicates the possibility of coumarin **4** as a potential chemosensor for Cu^2+^ ions. Figure 22 shows changes in the fluorescence intensity of 0.38 μM of 2-amino-4-hydroxy-6-(4-hydroxy-2-oxo-2H-chromen-3-yl)nicotinonitrile 4 with the increase in [Cu^2+^]. The quenching action of the fluorescence intensity by metal ions continued until the concentration of [Cu^2+^] 25 μM.

The Stern–Volmer relationship explores the kinetics of a photophysical quenching process of a studied dye caused by copper ions. The equation that is used to analyze the fluorescence quenching [67] is as follows:(2)$$\frac{I_{0}}{I}=1+K_{SV}\left[Q\right]$$

A linear plot of $\frac{I_{0}}{I}$ versus $\left[Q\right]$ indicates that the mechanism of fluorescence quenching could be called a dynamic diffusion process, where $\left[Q\right]\;$ is the quencher concentration (Cu^2+^), *I* and $I_{0}$ are the fluorescence intensities of 2-amino-4-hydroxy-6-(4-hydroxy-2-oxo-2H-chromen-3-yl)nicotinonitrile **4** in the presence and absence of Cu^2+^, respectively, and *K_SV_* is the Stern–Volmer quenching constant. This system is then likely to follow the Stern–Volmer relationship. The value of the Stern–Volmer quenching constant is calculated (slope) to be equal to 0.5 M^−1^ (Figure 24).

## 4. Conclusions

A series of coumarin derivatives **2**–**4** was synthesized by the multicomponent reaction procedure using 3-acetyl-4-hydroxycoumarin, 3-hydroxybenzaldehyde, malononitrile, or ethylcyanoacetate. This method offers several advantages, such as simplicity, short reaction time, ease of implementation, and high yield. The minimum inhibitory concentrations (MIC) of the synthesized compounds **2**–**4** were evaluated.The most active compound with broad spectrum inhibitory activity against *L. monocytogenes* is compound **3**. Furthermore, compound **3** exhibited cytotoxic activity equivalent to that of standard Vinblastine. On the other hand, the theoretical simulation showed that coumarin derivative **4** is an effective inhibitor of the tumor protein HCT-116. Based on these promising results, we can propose that these compounds be used in the formulation of antibiotics. The DFT study allowed us to investigate the reactivity and acidity of the d sites of the 2-amino-4-hydroxy-6-(4-hydroxy-2-oxo-2H-chromen-3-yl)nicotinonitrile **4** product. In addition, we found that the imine tautomer is more abundant than that of the alcohol. Moreover, the studied compound was found to be qualified for use as a sensor of pH, Cu^2+^ and Ni^2+^ ions due to its high sensitivity to fluorescence in acidic media.

Concerning the light properties, the occurrence of photon reabsorption is excluded, indicating promising chemical properties due to the high Stokes shift of the dye in most solvents. Furthermore, the UV-vis absorption spectra calculated by the TD-DFT approach are in good agreement with the experiment.

## Data Availability

Not applicable.

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
