# Peer review of "Chemosensing Properties of Coumarin Derivatives: Promising Agents with Diverse Pharmacological Properties, Docking and DFT Investigation"

_molecules, 2022, doi:10.3390/molecules27185921_

Round 1

Reviewer 1 Report (Previous Reviewer 3)

The current version of the manuscript is suitable for publication in Molecules. Compound 3 has now a good purity (according to the NMR spectra included in the Supplementary Information), so I recommend its publication in Molecules once the English style is revised.

Author Response

Dear editor,

Thank you very much for the comments, recommendations, and corrections

We have carefully checked the following items suggested by the editor, and we have re-arranged the main text according to the suggestions. All revisions have been given with the Yellow color.

We sincerely hope that you will find our revised manuscript suitable for publication as an origin article in your prestigious journal.

Looking forward to receiving a prompt reply,

Sincerely,

Pr. Naceur HAMDI

Reviewer 2 Report (New Reviewer)

Comments and suggestions for authors: 

The paper aims to describe the pharmacological proprieties of coumarin derivatives. Starting from a fast and easy reaction for the synthesis, the author presents several theoretical studies such as an explanation of the tautomeric equilibrium and a Mapped Electrostatic Potential Surface analysis to better explain the reaction trend and the proposed reaction pattern. Molecular Modeling studies are also performed to investigate the binding of compound 4 with several proteins. There are also UV calculations to study the absorption spectra of compound 4 in different solvents and fluorescent activity. Furthermore, the author presents biological tests on the synthesized compounds: antibacterial activity, antioxidant activity, and anti-inflammatory activity. In addition to that, compound 4 has also been explored for its ability to decrease fluorescence in the presence of certain ions, and thus exhibit chemosensing activity. 

The article resonates well because of the many theoretical and non-theoretical tests being performed by the authors. It also has a good impact as the compound exhibits highly desired chemosensor activity as an alarm in several diseases. 

The additional data with NMR results, UV and IR spectra are consistent with what is written and shown in the article.

The introduction should better explain the purpose of the work, especially regarding chemical sensors.

There should be a clearer naming of the compounds since in Scheme 1 compounds 2-3-4 are the initial and final compounds of the reaction, respectively, while in Scheme 3 the same numbers are used to describe intermediate compounds in the formation of compound 4 that do not correspond to the previous compounds. The use of the term "coumarin 4" is also unclear whether it refers to compound 4 or something else. I also suggest improving the clarity of the last sentence of the introduction where it is explained that one of the purposes of the work is to highlight the chemosensor ability of the synthesized compounds. 

In abstract line 26, the author should better specify the type of reaction because in these terms it is not easily understandable. 

In introduction line 55 the author should list the aims of the paper.

In the experimental section line 88-89/98-99-100/112 the Carom should be written as Carom.

In the experimental section line, 138 H2O2 should be written as H2O2.

In the experimental section line 207, there should be a better citation of the RCSB PDB as indicated in the following link https://www.rcsb.org/pages/policies.

In the experimental section line 238 the yield should be specified. 

In the result line 271, I suppose that the author intended to write “figure 3”.

Scheme 2 is missing, there is scheme 1 and scheme 3 but not scheme 2.

I would recommend more clarity in the exposition of theoretical data. In particular, it is not clear from the manuscript whether the UV calculations given on page 12 and figure 6 refer to something else than the UV calculations given on page 16 and figure 8, as the paragraph explanations and captions are the same. In fact, in line 335 the author writes “we have found that the UV-vis absorption theoretical spectra shape is similar to the experimental one (Fig. 8). But in line 369 the author writes “The UV absorption spectra were then computed for the coumarin 4 as a function of different solvents (water, ethanol, chloroform, acetonitrile, THF, and acetic acid). The obtained simulated spectra are presented in Fig.8.” 

Also, figure 8 and figure 6 have the same caption “Computed UV-abs spectra at TD-DFT B3LYP/6-31+G(d) level as a function of several solvents.” and it is not clear whether this test refers to two different compounds. I also would recommend more clarity regarding the citation of figures and tables and also recommend better schematization of the theoretical data part. 

I would recommend improving citations, particularly by including more recent articles.

Overall, the paper is very comprehensive and contains a wealth of well-explained tests and results consistent with what is stated as the purpose of the project but there should be an improvement in the exposition of the results, especially the theoretical ones.

Author Response

Comments and suggestions for authors: 

The paper aims to describe the pharmacological proprieties of coumarin derivatives. Starting from a fast and easy reaction for the synthesis, the author presents several theoretical studies such as an explanation of the tautomeric equilibrium and a Mapped Electrostatic Potential Surface analysis to better explain the reaction trend and the proposed reaction pattern. Molecular Modeling studies are also performed to investigate the binding of compound 4 with several proteins. There are also UV calculations to study the absorption spectra of compound 4 in different solvents and fluorescent activity. Furthermore, the author presents biological tests on the synthesized compounds: antibacterial activity, antioxidant activity, and anti-inflammatory activity. In addition to that, compound 4 has also been explored for its ability to decrease fluorescence in the presence of certain ions, and thus exhibit chemosensing activity. 

The article resonates well because of the many theoretical and non-theoretical tests being performed by the authors. It also has a good impact as the compound exhibits highly desired chemosensor activity as an alarm in several diseases. 

The additional data with NMR results, UV and IR spectra are consistent with what is written and shown in the article.

The introduction should better explain the purpose of the work, especially regarding chemical sensors.

There should be a clearer naming of the compounds since in Scheme 1 compounds 2-3-4 are the initial and final compounds of the reaction, respectively, while in Scheme 3 the same numbers are used to describe intermediate compounds in the formation of compound 4 that do not correspond to the previous compounds. The use of the term "coumarin 4" is also unclear whether it refers to compound 4 or something else. I also suggest improving the clarity of the last sentence of the introduction where it is explained that one of the purposes of the work is to highlight the chemosensor ability of the synthesized compounds. 

In abstract line 26, the author should better specify the type of reaction because in these terms it is not easily understandable. 

In introduction line 55 the author should list the aims of the paper.

In the experimental section line 88-89/98-99-100/112 the Carom should be written as Carom.

In the experimental section line, 138 H2O2 should be written as H2O2.

In the experimental section line 207, there should be a better citation of the RCSB PDB as indicated in the following link https://www.rcsb.org/pages/policies.

In the experimental section line 238 the yield should be specified. 

In the result line 271, I suppose that the author intended to write “figure 3”.

Scheme 2 is missing, there is scheme 1 and scheme 3 but not scheme 2.

Remark: I would recommend more clarity in the exposition of theoretical data. In particular, it is not clear from the manuscript whether the UV calculations given on page 12 and figure 6 refer to something else than the UV calculations given on page 16 and figure 8, as the paragraph explanations and captions are the same. In fact, in line 335 the author writes “we have found that the UV-vis absorption theoretical spectra shape is similar to the experimental one (Fig. 8). But in line 369 the author writes “The UV absorption spectra were then computed for the coumarin 4 as a function of different solvents (water, ethanol, chloroform, acetonitrile, THF, and acetic acid). The obtained simulated spectra are presented in Fig.8.” 

Also, figure 8 and figure 6 have the same caption “Computed UV-abs spectra at TD-DFT B3LYP/6-31+G(d) level as a function of several solvents.” and it is not clear whether this test refers to two different compounds. I also would recommend more clarity regarding the citation of figures and tables and also recommend better schematization of the theoretical data part. 

I would recommend improving citations, particularly by including more recent articles.

Overall, the paper is very comprehensive and contains a wealth of well-explained tests and results consistent with what is stated as the purpose of the project but there should be an improvement in the exposition of the results, especially the theoretical ones.

Answer: Thank you for your comments. We have taken into account your remarks and modified the TD-DFT analysis part and, generally the theoretical section. We have revised all tables and figures captions.

The authors of “Chemosensing Properties of Coumarin Derivatives: Promising 2 Agents with Diverse Pharmacological Properties, Docking and 3 DFT Investigation” made an interesting work. Nevertheless, there are some points that need to be treated before this work can be publish.

Remark: They calculated different global and local chemical reactivity descriptors of the compounds. Nevertheless, they do not discuss the values of all these descriptors or the meaning of their calculation. I recommend the following works as examples that how it can be exploited these descriptors: https://doi.org/10.2478/s11532-014-0555-x; https://doi.org/10.1016/j.bmc.2018.10.045; https://doi.org/10.1016/j.molstruc.2020.128456.

Answer: Thank you for your remark. Indeed, we have modified the computational details part. The proposed mechanism doesn’t display a regioselectivity issue. We just investigated the maximum and minimum potential values, which deduced from the MESP surfaces in order to analyze the reactivity (mobility) of the protons in the one hand, and the nucleophilicity of the heteroatoms(O and N) on the other hand.

Remark: In the same manner, they performed molecular docking over several biological targets. Nevertheless, they never justify why they choose those proteins.

Answer: Thank you for your remark. Indeed, in the “docking simulations” part, the objective wasn’t to confirm the biological activities of the synthesized organic product, which have been proven experimentally. The docking part offers us the advantage of studying in detail the interactions between the organic molecule and the amino acids of proteins. The docking part allows us to analyze in depth the functions and sites responsible for such biological activity. So the contribution of this theoretical study is to find explanations for experimental results and the choice of proteins fell on a bibliographic study and previous work on the activities of similar organic molecules with PLA2, lipoxygenase, HepG-2 and HCT-116.

Reviewer 3 Report (New Reviewer)

The authors of “Chemosensing Properties of Coumarin Derivatives: Promising 2 Agents with Diverse Pharmacological Properties, Docking and 3 DFT Investigation” made an interesting work. Nevertheless, there are some points that need to be treated before this work can be publish.

They calculated different global and local chemical reactivity descriptors of the compounds. Nevertheless, they do not discuss the values of all these descriptors or the meaning of their calculation. I recommend the following works as examples that how it can be exploited these descriptors: https://doi.org/10.2478/s11532-014-0555-x; https://doi.org/10.1016/j.bmc.2018.10.045; https://doi.org/10.1016/j.molstruc.2020.128456.

In the same manner, they performed molecular docking over several biological targets. Nevertheless, they never justify why they choose those proteins.

Author Response

We have added these references and we justify the use of proteines, please check the revised paper 

This manuscript is a resubmission of an earlier submission. The following is a list of the peer review reports and author responses from that submission.

Round 1

Reviewer 1 Report

The manuscript describes the synthesis and chemosensing properties of coumarin derivatives. The manuscript is not publishable in its current form due to following major issues.

1)      Lack of focus. Initially the authors screened their compounds against various microorganisms, then carried out antioxidant assays, subsequently tested as anti-inflammatory, then tested in cancer cell lines.

2)      Some of the docking studies are not backed with in vitro biological target data.

3)      Computational part is overdeveloped.

4)      Too many figures and tables that can easily go into supporting information.

5)      Table 12 and 13: no errors shown for IC50 values.

6)      There are better LOX inhibitors to be used as positive controls than indomethacin which is primarily a COX inhibitor.

7)      Table 14 is confusing. What is HepG-2 and pdb. HepG-2 is a cell line not a target protein. Same applies for HCT-116.

8)      Because of unfocused nature of this paper, it loses its value for publication.

Author Response

1)      Lack of focus. Initially the authors screened their compounds against various microorganisms, then carried out antioxidant assays, subsequently tested as anti-inflammatory, then tested in cancer cell lines.

correction was done, you are requested to check the revised paper

2)      Some of the docking studies are not backed with in vitro biological target data.

coorection was done 

3)      Computational part is overdeveloped.

we have revised the Computational  study

4)      Too many figures and tables that can easily go into supporting information.

we have moved some figures and tables to supporting information 

5)      Table 12 and 13: no errors shown for IC50 values.

correction was done 

6)      There are better LOX inhibitors to be used as positive controls than indomethacin which is primarily a COX inhibitor.

its a good remark we will take with this in our next work 

7)      Table 14 is confusing. What is HepG-2 and pdb. HepG-2 is a cell line not a target protein. Same applies for HCT-116.

correction was done

Reviewer 2 Report

Al-Hazmy et al studied coumarin derivatives using many many different techniques. The have synthesized the compounds (organic chemistry), studied them using theoretical means (quantum chemistry and docking) and experimental methods, performed biochemical assays. It is about impossible for a single reviewer to judge the quality of the findings from each method.

In any case, the paper is written in a reasonably good English and logically built by showing the results obtained with the various methods and there is a clear aim to show which compounds could be the most effective. However, I find the paper a too long, some data might be moved to the SI.  

I have some suggestions:

(1) I appreciate that the authors used conceptual DFT to study there compounds, but I find that they should be more careful at least in some places with the interpretation of their data. E.g. they write "the four hydrogens of benzene are the more acidic ones since they possess the highest and" . Even if they find the smallest/largest Fukui function for these atoms, they should seriously think about the acidity of the hydrogen atoms of the phenyl ring. What ever the index says, these atoms are not acidic!!!

(2) They write: "The tautomeric equilibrium (Imine-Amine) as indicated in Fig. 3 was studied." They should prepare a much better scheme or figure to show what the exact transformation is, where the proton is moving. The obtained energy of activation is by far too high, they should consider recalculating the barrier by implicitly including solvent molecule (e.g.) water to facilitate the proton transfer within the molecule. Furthermore, they could cite relevant data from literature and compare the obtained value with this. 

(4) This sentence is not good: "The of the UV-vis absorption theoretical spectra (Fig. 22) are in good agreement with the experimental ones (Fig. 6) ." It is not correct and I think Fig.22 is referred to by accident. 

(5) "We have found that THF and acetic acid give the lowest values. At another level, we have predicted the fluorescence quantum yield () by using spectral analysis software." What does "at another level" mean? 

(6) the authors draw many many trendlines but do not put any R2 values to them.. maybe they are too small and show no significant correlation?

(7) "In Figures 11. It was noticed from the absorption spectra, that there is a relative difference"  Did they only mean figure 11?

The paper also contains minor mistakes e.g. 

(1) the parentheses are not all ok in equation 4 and 5

(2) in chapter 2.13 there is the whole reference which should not appear there

(3) this sentence is not really OK (around line 260): 

"Due to the exceptional reactivity of the acetyl group in 3-acetyl-4-hydroxycoumarin as well as the versatile biological activities of coumarin derivatives," 

(4) it is difficult or impossible to see atom numbering above Table 2

Overall the paper is reasonably well-written and if the authors did all these studies, the results merit publication. 

Author Response

Remark 1

  • I appreciate that the authors used conceptual DFT to study there compounds, but I find that they should be more careful at least in some places with the interpretation of their data. E.g. they write "the four hydrogens of benzene are the more acidic ones since they possess the highest and" . Even if they find the smallest/largest Fukui function for these atoms, they should seriously think about the acidity of the hydrogen atoms of the phenyl ring. What ever the index says, these atoms are not acidic!!! (acidité de protons de phenyls)

Reply 1

            Thank you for your comment. We have removed the discussion about the acidity of phenyl protons. Also, we have modified the manuscript and we have only discussed the MESP surfaces (page 17) in order to investigate the reactivity selected sites and check the proposed mechanism.

  • They write: "The tautomeric equilibrium (Imine-Amine) as indicated in Fig. 3 was studied." They should prepare a much better scheme or figure to show what the exact transformation is, where the proton is moving. The obtained energy of activation is by far too high, they should consider recalculating the barrier by implicitly including solvent molecule (e.g.) water to facilitate the proton transfer within the molecule. Furthermore, they could cite relevant data from literature and compare the obtained value with this.

Reply 2

           Thank you for your relevant comment. We have taken into account your remark and we have studied the impact of water as solvent on the activation energy and, the stability of the predicted tautomers. In this investigation we have compared the implicit and explicit solvatation models. We have found that only the latter leads to reduce the activation energy value.

(4) This sentence is not good: "The of the UV-vis absorption theoretical spectra (Fig. 22) are in good agreement with the experimental ones (Fig. 6)." It is not correct and I think Fig.22 is referred to by accident.

Reply 4

Thank you for your remark; we have verified and corrected the Table label. Also, we have improved this paragraph with taken into account your suggestion.

(5) "We have found that THF and acetic acid give the lowest values. At another level, we have predicted the fluorescence quantum yield () by using spectral analysis software." What does "at another level" mean?

Thank you for your remark. In the manuscript, the word “at another level” is replaced by “Also”. The latest is suitable in the context of the paragraph.

(6) the authors draw many many trendlines but do not put any R2 values to them.. maybe they are too small and show no significant correlation?

(7) "In Figures 11. It was noticed from the absorption spectra, that there is a relative difference"  Did they only mean figure 11?

Reviewer 3 Report

This work is a thorough research on coumarin sensors that is worthy publishable in Molecules, but there are some points that the authors should address in any subsequent revision:

1) A supporting information including all the spectra (NMR, IR and HRMS) of the synthesized compounds must be provided. The authors wrote in the abstract that they characterize the products by mass spectrometry, but no data of HRMS are provided in the manuscript. They should include this in the main text.

2) Line 104: the authors wrote DMC. I suppose they wanted to write DCM (dichloromethane), but they should check this issue and include the corresponding abbreviation in the manuscript.

3) Even if they did similar studies in previous works, the authors should describe in depth subsections 2.5, 2.8, 2.9 and 2.13.

4) There a lot of format issues in the manuscript (missing dots, numbers of products without bold style, different styles in references...).

Author Response

This work is a thorough research on coumarin sensors that is worthy publishable in Molecules, but there are some points that the authors should address in any subsequent revision:

1) A supporting information including all the spectra (NMR, IR and HRMS) of the synthesized compounds must be provided. The authors wrote in the abstract that they characterize the products by mass spectrometry, but no data of HRMS are provided in the manuscript. They should include this in the main text.

please check the attached doc in the revised paper 

2) Line 104: the authors wrote DMC. I suppose they wanted to write DCM (dichloromethane), but they should check this issue and include the corresponding abbreviation in the manuscript.

no its DMC: dimethylcarbonate

3) Even if they did similar studies in previous works, the authors should describe in depth subsections 2.5, 2.8, 2.9 and 2.13.

corrections were done , please check the revised paper 

4) There a lot of format issues in the manuscript (missing dots, numbers of products without bold style, different styles in references...).

we have improved the manuscript as much as we can 

Reviewer 4 Report

The manuscript provides a comprehensive research of novel coumarin derivatives. In my opinion the manuscript should be accepted after minor revision.

1. The abstract and title should be shorter.

2. SI with NMR plots should be added.

3. Lines 96, 98, 117: what is d ppm?

4. Line 97: coupling constants need editing.

5. Line 99, 108, 119: each signal in 13C NMR should be listed instead of “115.0–136.0 (Carom).”.

6. Elemental analyses should be added to the section 2.

7. Line 143: what extract?

8. Line 296: what molecule?

Author Response

Comments and Suggestions for Authors

The manuscript provides a comprehensive research of novel coumarin derivatives. In my opinion the manuscript should be accepted after minor revision.

  1. The abstract and title should be shorter.

      the abstract and the title were shorter 

  1. SI with NMR plots should be added.

  The plot NMR was added

  1. Lines 96, 98, 117: what is d ppm?

 correction was done 

  1. Line 97: coupling constants need editing.

 correction was done 

  1. Line 99, 108, 119: each signal in 13C NMR should be listed instead of “115.0–136.0 (Carom).”.

 correction was done 

  1. Elemental analyses should be added to the section 2.

 elemental analysis was added 

  1. Line 143: what extract?

 correction was done 

  1. Line 296: what molecule?

 correction was done 

Round 2

Reviewer 1 Report

The manuscript is not suitable for publication in any journal. The manuscript still has many flaws:

1)      RCSB acronym is incorrectly expanded as Royal Col laboratory for Structural Bioinformatics.

2)      Section 3. Scheme 1 phrase just appear randomly instead of in the bracket where it belongs.

3)      It is unreasonable to write so much detail about spectral data of a simple and widely known molecule, 3-acetyl-4-hydroxycoumarin.

4)      Unfortunately, the manuscript is still heavy on computational part. Many pieces of computational data remained dissociated from actual biological properties of the synthesized molecules.

5)      Presentation is very poor. For example, Table 9. were reported for -----. Table 9 title fungs. Likewise entire manuscript has below standard level of presentation.

6)      Docking studies has no in vitro data that synthesized compounds inhibit those targets LOX, PLA2, and it is not clear what is the name of the protein for PDBs used under antiproliferative section in Table 14 and 15.

7)      Computational studies are separate pieces of the manuscript that are not tied with observed experimental data.

8)      Table 12: IC50 Errors are still missing. The authors state that PLA2 IC50s for compounds 2-4 ranges from 26.5 uM -25.4 uM but table 12 shows IC50 values >1000 uM.   

Reviewer 3 Report

After thorough evaluation of the revised version, I think that the paper can not be published in Molecules because of the purity of the main compound 3. NMR spectra of compound 3 have a lot of signals related to impurities. 

The authors should purify this compound and provide clean NMR spectra. Interestingly, this compound is the one showing the best inhibitory activity. Is it because of the compound itself or because of the impurities? Once they have the pure product, they should repeat all the experiments with this product in order to check the validity of their work. 

It is curious that with this considerable amount of impurities, the theoretical and experimental Elemental Analysis almost is the same: Anal. Calcd for C21H12O5N2 : 103 C, 67.74 %; H 3.25 %; N, 7.52%. Found: C, 67.8; H, 3.25; N,7.5.

As a minor points, the numbers of the chemical formulas should be included as subindex. 

I would recommend rejecting the article so that the authors have enough time to solve this issue and, once they have adequately completed their research, resubmit it.